# Confounding-Robust Policy Improvement

**Nathan Kallus**
Cornell University and Cornell Tech
New York, NY
kallus@cornell.edu

**Angela Zhou**
Cornell University and Cornell Tech
New York, NY
az434@cornell.edu

## Abstract

We study the problem of learning personalized decision policies from observational data while accounting for possible unobserved confounding in the data-generating process. Unlike previous approaches that assume unconfoundedness, *i.e.*, no unobserved confounders affected both treatment assignment and outcomes, we calibrate policy learning for realistic violations of this unverifiable assumption with uncertainty sets motivated by sensitivity analysis in causal inference. Our framework for confounding-robust policy improvement optimizes the minimax regret of a candidate policy against a baseline or reference "status quo" policy, over an uncertainty set around nominal propensity weights. We prove that if the uncertainty set is well-specified, robust policy learning can do no worse than the baseline, and only improve if the data supports it. We characterize the adversarial subproblem and use efficient algorithmic solutions to optimize over parametrized spaces of decision policies such as logistic treatment assignment. We assess our methods on synthetic data and a large clinical trial, demonstrating that confounded selection can hinder policy learning and lead to unwarranted harm, while our robust approach guarantees safety and focuses on well-evidenced improvement.

## 1 Introduction

The problem of learning personalized decision policies to study "what works and for whom" in areas such as medicine and e-commerce often endeavors to draw insights from observational data, since data from randomized experiments may be scarce and costly or unethical to acquire [12, 3, 30, 6, 13]. These and other approaches for drawing conclusions from observational data in the Neyman-Rubin potential outcomes framework generally appeal to methodologies such as inverse-propensity weighting, matching, and balancing, which compare outcomes across groups constructed such that assignment is almost as if at random [23]. These methods rely on the controversial assumption of *unconfoundedness*, which requires that the data are sufficiently informative of treatment assignment such that no unobserved confounders jointly affect treatment assignment and individual response [24]. This key assumption may be made to hold *ex ante* by directly controlling the treatment assignment policy as sometimes done in online advertising [4], but in other domains of key interest such as personalized medicine where electronic medical records (EMRs) are increasingly being analyzed *ex post*, unconfoundedness may never truly hold in fact.

Assuming unconfoundedness, also called *ignorability*, *conditional exogeneity*, or *selection on observables*, is controversial because it is fundamentally unverifiable since the counterfactual distribution is not identified from the data, thus rendering any insights from observational studies vulnerable to this fundamental critique [11]. If the data is truly unconfounded, it would be known by construction because it would come from an RCT or logged bandit; any data whose unconfoundedness is uncertain must be confounded to *some* extent. The growing availability of richer observational data such as found in EMRs renders unconfoundedness more plausible, yet it still may never be fully satisfied in practice. Because unconfoundedness may fail to hold, existing policy learning methods that assume it can lead to personalized decision policies that seek to exploit individual-level effects that are not

really there, may intervene where not necessary, and may in fact lead to net harm rather than net good. Such dangers constitute obvious impediments to the use of policy learning to enhance decision making in such sensitive applications as medicine, public policy, and civics.

To address this deficiency, in this paper we develop a framework for robust policy learning and improvement that can ensure that a personalized decision policy derived from observational data, which inevitably may have some unobserved confounding, does no worse than a current policy such as the standard of care and in fact does better if the data can indeed support it. We do so by recognizing and accounting for the potential confounding in the data and require that the learned policy improve upon a baseline no matter the direction of confounding. Thus, we calibrate personalized decision policies to address sensitivity to realistic violations of the unconfoundedness assumption. For the purposes of informing reliable and personalized decision-making that leverages modern machine learning, point identification of individual-level causal effects, which previous approaches rely on, may not be at all necessary for success, but accounting for the lack of identification is.

Functionally, our approach is to optimize a policy to achieve the best worst-case improvement relative to a baseline treatment assignment policy such as treat all or treat none, where the improvement is measured using a weighted average of outcomes and weights take values in an uncertainty set around the nominal inverse propensity weights (IPW). This generalizes the popular class of IPW-based approaches to policy learning, which optimize an unbiased estimator for policy value under unconfoundedness [15, 28, 27]. Unlike standard approaches, in our approach the choice of baseline is material and changes the resulting policy chosen by our method. This framing supports reliable decision-making in practice, as often a practitioner is seeking evidence of substantial improvement upon the standard of care or a default option, and/or the intervention under consideration introduces risk of toxicity or adverse effects and should not be applied without strong evidence.

Our contributions are as follows: we provide a framework for performing *policy improvement* which is robust in the face of unobserved confounding. Our framework allows for the specification of data-driven uncertainty sets, based on the sensitivity parameter $\Gamma$ describing a pointwise multiplicative bound, as well as allowing for a global uncertainty budget which restricts the total deviation proportionally to the maximal $\ell_1$ discrepancy between the true propensities and nominal propensities. Leveraging the optimization structure of the robust subproblem, we provide algorithms for performing policy optimization. We assess performance on a synthetic example as well as a large clinical trial.

## 2   Problem Statement and Preliminaries

We assume the observational data consists of tuples of random variables $\{(X_i, T_i, Y_i) : i = 1, \ldots, n\}$, comprising of covariates $X_i \in \mathcal{X}$, assigned treatment $T_i \in \{-1, 1\}$, and real-valued outcomes $Y_i \in \mathbb{R}$. Using the Neyman-Rubin potential outcomes framework, we let $Y_i(-1)$ and $Y_i(1)$ denote the potential outcomes of applying treatment $-1$ and $1$, respectively. We assume that the observed outcome is potential outcome for the observed treatment, $Y_i = Y_i(T_i)$, encapsulating non-interference and consistency, also known as SUTVA [25]. We also use the convention that the outcomes $Y_i$ corresponds to losses so that lower outcomes are better.

We consider evaluating and learning a (randomized) treatment assignment policy mapping covariates to the probability of assigining treatment, $\pi : \mathcal{X} \to [0, 1]$. We focus on a policy class $\pi \in \mathcal{F}$ of restricted complexity. Examples include linear policies $\pi_\beta(X) = \mathbb{I}[\beta^\mathsf{T} x]$, logistic policies $\pi_\beta(X) = \sigma(\beta^\mathsf{T} x)$ where $\sigma(z) = 1/(1 + e^{-z})$, or decision trees of a bounded certain depth. We allow the candidate policy $\pi$ to be either deterministic or stochastic, and denote the random variable indicating the realization of treatment assignment for some $X_i$ to be a Bernoulli random variable $Z_i^\pi$ such that $\pi(X_i) = \Pr[Z_i^\pi = 1 \mid X_i]$.

The goal of policy evaluation is to assess the *policy value*,

$$V(\pi) = \mathbb{E}[Y(Z^\pi)] = \mathbb{E}[\pi(X_i)Y(1) + (1 - \pi(X_i))Y(-1)],$$

the population average outcome induced by the policy $\pi$. The problem of *policy optimization* seeks to find the best such policy over the parametrized function class $\mathcal{F}$. Both of these tasks are hindered by residual confounding since then $V(\pi)$ cannot actually be identified from the data.

Motivated by the sensitivity model in [22] and without loss of generality, we assume that there is an additional but unobserved covariate $U_i$ such that unconfoundedness would hold if we were to control for both $X_i$ *and* $U_i$, that is, such that $\mathbb{E}[Y_i(t) \mid X_i, U_i, T_i] = \mathbb{E}[Y_i(t) \mid X_i, U_i]$ for $t \in \{-1, 1\}$.

Equivalently, we can treat the data as collected under an unknown logging policy that based its assignment on both $X_i$ *and* $U_i$ and that assigned $T_i = 1$ with probability $e(X_i, U_i) = \Pr[T = 1 \mid X_i, U_i]$. Here, $e(X_i, U_i)$ is precisely the *true* propensity score of unit $i$. Since we do not have access to $U_i$ in our data, we instead presume that we have access only to *nominal* propensities $\hat{e}(X_i) = \Pr[T = 1 \mid X_i]$, which *do not* account for the potential unobserved confounding. These are either part of the data or can be estimated directly from the data using a probabilistic classification model such as logistic regression. For compactness, we denote $\hat{e}_{T_i}(X_i) = \frac{1}{2}(1 + T_i)\hat{e}(X_i) + \frac{1}{2}(1 - T_i)(1 - \hat{e}(X_i))$ and $e_{T_i}(X_i, U_i) = \frac{1}{2}(1 + T_i)e(X_i, U_i) + \frac{1}{2}(1 - T_i)(1 - e(X_i, U_i))$.

## 2.1 Related Work

Our work builds upon the literatures on policy learning from observational data and on sensitivity analysis in causal inference.

**Sensitivity analysis.** Sensitivity analysis in causal inference tests the robustness of qualitative conclusions made from observational data to model specification or assumptions such as unconfoundedness. In this work, we focus on structural assumptions bounding how unobserved confounding affects selection, without restriction on how unobserved confounding affects outcomes. In particular, we focus on the implications of confounding on personalized treatment decisions.

Rosenbaum's model for sensitivity analysis assesses the robustness of matched-pairs randomization inference to the presence of unobserved confounding by considering a uniform bound $\Gamma$ on the impact of confounding on the *odds ratio* of treatment assignment [22]. Motivated by a logistic specification, in this model, the odds-ratio for two units with the same covariates $X_i = X_j$, which differs due to the units' different values $U_i, U_j$ for the unobserved confounder, is $e^{\log(\Gamma)(U_i - U_j)}$, and $U_i, U_j \in [0, 1]$ may be arbitrary. We consider a variant, also called the "marginal sensitivity model" in [34], which instead bounds the log-odds ratio between $e(X_i), e(X_i, U_i)$.

In the sampling literature, the weight-normalized estimator for population mean is known as the Hajek estimator, and Aronow and Lee [1] derive sharp bounds on the estimator arising from a uniform bound on the sampling weights, showing a closed-form solution for the solution to the fractional linear program for a *uniform* bound on the sampling probabilities. [34] considers bounds on the Hajek estimator, but imposes a parametric model on the treatment assignment probability.

Sensitivity analysis is also related to the literature on partial identification of treatment effects [17]. Similar bounds studied in [33] in the transfer learning setting rely on no knowledge but the law of total probability. Our approach instead uses sensitivity analysis based on the estimated propensities as a starting point and leverages additional information about how far it is from true propensities to achieve tighter bounds that interpolate between the fully-unconfounded and arbitrarily-confounded regimes. [19] considers tightening the bounds from the Hajek estimator by adding shape constraints, such as log-concavity, on the cumulative distribution of outcomes $Y$. [18] considers sharp partially identified bounds under the assumption of an uniform bound on nominal propensities, $\sup_U |\Pr[T = 1 \mid X] - \Pr[T = 1 \mid X, U]| \le c$. We focus on the implications of sensitivity analysis for policy-learning based approaches for learning optimal treatment policies from observational data.

**Policy learning from observational data under unconfoundedness.** A variety of approaches for learning personalized intervention policies that maximize causal effect have been proposed, but all under the assumption of unconfoundedness. These fall under regression-based strategies [21] or reweighting-based strategies [3, 12, 13, 28], or doubly robust combinations thereof [6, 30]. Regression-based strategies estimate the conditional average treatment effect (CATE), $\mathbb{E}[Y(1) - Y(-1) \mid X]$, either directly or by differencing two regressions, and use it to score the policy. Without unconfoundedness, however, CATE is not identifiable from the data and these methods have no guarantees.

Reweighting-based strategies use inverse-probability weighting (IPW) to change measure from the outcome distribution induced by a logging policy to that induced by the policy $\pi$. Specifically, these methods use the fact that, under unconfoundedness, $\hat{V}^{\text{IPW}}(\pi)$ is unbiased for $V(\pi)$ [15], where

$$\hat{V}^{\text{IPW}}(\pi) = \frac{1}{n} \sum_{i=1}^n \frac{(1 + T_i(2\pi(X_i) - 1))Y_i}{2\hat{e}_{T_i}(X_i)} \tag{1}$$

Optimizing $\hat{V}^{\text{IPW}}(\pi)$ can be phrased as a weighted classification problem [3]. Since dividing by propensities can lead to extreme weights and high variance estimates, additional strategies such as

clipping the probabilities away from 0 and normalizing by the sum of weights as a control variate are typically necessary for good performance [27, 32]. With or without these fixes, if there are unobserved confounders, none of these are consistent for $V(\pi)$ and learned policies may introduce more harm than good.

A separate literature in reinforcement learning considers the idea of safe policy improvement by minimizing the regret against a baseline policy, forming an uncertainty set around the presumed unknown transition probabilities between states as in [29], or forming a trust region for safe policy exploration via concentration inequalities on the importance-reweighted estimates of policy risk [20].

## 3 Robust policy evaluation and improvement

Our framework for confounding-robust policy improvement minimizes a bound on policy regret against a specified baseline policy $\pi_0$, $R_{\pi_0}(\pi) = V(\pi) - V(\pi_0)$. Our bound is achieved by maximizing a reweighting-based regret estimate over an uncertainty set around the nominal propensities. This ensures that we cannot do any worse than $\pi_0$ and may do better, even if the data is confounded.

The baseline policy $\pi_0$ can be any fixed policy that we want to make sure not to do worse than, or deviate from unnecessarily. This is usually the current standard of care, established from prior evidence, and can be a policy that actually depends on $x$. Generally, we think of this as the policy that always assigns control. Alternatively, if a reliable estimate of the average treatment effect, $\mathbb{E}[Y(1) - Y(-1)]$, is available then $\pi_0$ can be the constant $\pi_0(x) = \mathbb{I}[\mathbb{E}[Y(1) - Y(-1)] < 0]$. In an agnostic extreme, $\pi_0$ can be the complete randomization policy $\pi_0(x) = 1/2$.

### 3.1 Confounding-robust policy learning by optimizing minimax regret

If we had oracle access to the true inverse propensities $W_i^* = 1/e_{T_i}(X_i, U_i)$ we could form the correct IPW estimate by replacing nominal with true propensities in eq. (1). We may go a step further and, recognizing that $\mathbb{E}[1/e_{T_i}(X_i, U_i)] = 2$, use the empirical sum of true propensities as a control variate by normalizing our IPW estimate by them. This gives rise to the following Hajek estimators of $V(\pi)$ and correspondingly $R_{\pi_0}(\pi)$

$$\hat{V}^*(\pi) = \frac{\sum_{i=1}^{n} W_i^*(1 + T_i(2\pi(X_i) - 1))Y_i}{\sum_{i=1}^{n} W_i^*},$$

$$\hat{R}_{\pi_0}^*(\pi) = \hat{V}^*(\pi) - \hat{V}^*(\pi_0) = \frac{2\sum_{i=1}^{n} W_i^*(\pi(X_i) - \pi_0(X_i))T_i Y_i}{\sum_{i=1}^{n} W_i^*}$$

It follows by Slutsky's theorem that these estimates remain consistent (if we know $W_i^*$). Note that had we known $W_i^*$, both the normalization and choice of $\pi_0$ would have amounted to constant shifts and scales to $\hat{R}_{\pi_0}^*(\pi)$ that would not have changed the choice of $\pi$ to minimize the regret estimate. This will not be true of our bound, where both the normalization and the choice of $\pi_0$ will be material.

Since the oracle weights $W_i^*$ are unknown, we instead minimize the worst-case possible value of our regret estimate, by ranging over the space of possible values for $e_{T_i}(X_i, U_i)$ that are consistent with the observed data and our assumptions about the confounded data-generating process. Specifically, our model restricts the extent to which unobserved confounding may affect assignment probabilities. We first consider an uncertainty set motivated by the odds-ratio characterization in [22], which restricts how far the weights can vary pointwise from the nominal propensities. Given a bound $\Gamma > 1$, the odds-ratio restriction on $e(x, u)$ is that it satisfy the following inequalities

$$\Gamma^{-1} \leq \frac{(1 - \hat{e}(x))e(x, u)}{\hat{e}(x)(1 - e(x, u))} \leq \Gamma. \tag{2}$$

This restriction is motivated by (but more general than) considering a logistic model where $e(x, u) = \sigma(g(x) + \gamma u)$, $g$ is any function, $u \in [0, 1]$ is bounded without loss of generality, and $|\gamma| \leq \log(\Gamma)$. Such a model would necessarily give rise to eq. (2). This restriction also immediately leads to an uncertainty set for the true inverse propensities of observed treatments of each unit, $1/e(X_i, U_i)$, which we denote as follows

$$\mathcal{U}_n^{\Gamma} = \left\{ W \in \mathbb{R}_+^n : a_i^{\Gamma} \leq W_i \leq b_i^{\Gamma} \ \forall i = 1, \ldots, n \right\}, \quad \text{where}$$

$$a_i^{\Gamma} = \frac{1 - \hat{e}_{T_i}(X_i) + \Gamma\hat{e}_{T_i}(X_i)}{\Gamma\hat{e}_{T_i}(X_i)}, \ b_i^{\Gamma} = \frac{\Gamma(1 - \hat{e}_{T_i}(X_i)) + \hat{e}_{T_i}(X_i)}{\hat{e}_{T_i}(X_i)}$$

The corresponding bound on empirical regret is $\overline{R}_{\pi_0}(\pi; \mathcal{U}_n^\Gamma)$, where for any $\mathcal{U} \subset \mathbb{R}_+^n$ we define

$$\overline{R}_{\pi_0}(\pi; \mathcal{U}) = \sup_{W \in \mathcal{U}} \frac{2 \sum_{i=1}^n W_i(\pi(X_i) - \pi_0(X_i)) T_i Y_i}{\sum_{i=1}^n W_i}$$

We then choose the policy $\pi$ in our class that minimizes this regret bound, *i.e.*, $\overline{\pi}(\mathcal{F}, \mathcal{U}_n^\Gamma, \pi_0)$, where

$$\overline{\pi}(\mathcal{F}, \mathcal{U}, \pi_0) \in \operatorname{argmin}_{\pi \in \mathcal{F}} \overline{R}_{\pi_0}(\pi; \mathcal{U}) \tag{3}$$

In particular, for our estimate $\overline{R}_{\pi_0}(\pi; \mathcal{U}_n^\Gamma)$, weight normalization is crucial for only enforcing robustness against consequential realizations of confounding which affect the *relative* weighting of patient outcomes; otherwise robustness against confounding would simply assign weights to their highest possible bounds for positive $Y_i T_i$. If the baseline policy is in the policy class $\mathcal{F}$, it already achieves 0 regret; thus, minimizing regret necessitates learning regions of policy treatment assignment where evidence from observed outcomes suggests benefits in terms of decreased loss. Different baseline policies $\pi_0 = 0, 1$ structurally change the solution to the adversarial subproblem by shifting the contribution of the loss term $Y_i T_i(\pi(X_i) - \pi_0)$ to emphasize improvement upon the baseline.

**Budgeted uncertainty sets to address "local" confounding.** Our approach can be pessimistic in ensuring robustness against worst-case realizations of unobserved confounding "globally" for each unit, whereas concerns about unobserved confounding may be restricted to a subset of the population, due to subgroup risk factors or outliers. For the Rosenbaum model in hypothesis testing, this has been recognized by [7, 9] who address it by limiting the average of the unobserved propensities by an additional sensitivity parameter. Motivated by this, we next consider an alternative uncertainty set, where we fix a budget $\Lambda$ for how much the weights can diverge from the nominal inverse propensity weights in total. Specifically, letting $\hat{W}_i = 1/\hat{e}_{T_i}(X_i)$, we construct the uncertainty set

$$\mathcal{U}_n^{\Gamma, \Lambda} = \left\{ W \in \mathbb{R}_+^n : \sum_{i=1}^n |W_i - \hat{W}_i| \le \Lambda, \ a_i^\Gamma \le W_i \le b_i^\Gamma \ \forall i = 1, \ldots, n \right\}$$

When plugged into eq. (3), this provides an alternative policy choice criterion that is less conservative. We suggest to calibrate $\Lambda$ as a fraction $\rho < 1$ of the total deviation allowed by $\mathcal{U}_n^\Gamma$. Specifically, $\Lambda = \rho \sum_{i=1}^n \max(\hat{W}_i - a_i^\Gamma, b_i^\Gamma - \hat{W}_i)$. This is the approach we take in our empirical investigation.

### 3.2 The Improvement Guarantee

We next prove that if we appropriately bounded the potential hidden confounding then our worst-case empirical regret objective is asymptotically an upper bound on the true population regret. On the one hand, since our objective is necessarily non-positive if $\pi_0 \in \mathcal{F}$, this says we do no worse. On the other hand, if our objective is negative, which we can check by just evaluating it, then we are assured some strict improvement. Our result is generic for both $\mathcal{U}_n^\Gamma$ and $\mathcal{U}_n^{\Gamma, \Lambda}$.

Our upper bound depends on the complexity of our policy class. Define its Rademacher complexity:

$$\mathfrak{R}_n(\mathcal{F}) = \frac{1}{2^n} \sum_{\epsilon \in \{-1, +1\}^n} \sup_{\pi \in \mathcal{F}} \left| \frac{1}{n} \sum_{i=1}^n \epsilon_i \pi(X_i) \right|$$

All the policy classes we consider have $\sqrt{n}$-vanishing complexities, *i.e.*, $\mathfrak{R}_n(\mathcal{F}) = O(n^{-1/2})$.

**Theorem 1.** *Suppose that $(1/e(X_1, U_1), \ldots, 1/e(X_n, U_n)) \in \mathcal{U}$ and that $\nu \le e(x, u) \le 1 - \nu$ for some $\nu > 0$ and $|Y| \le C$ for some $C \ge 1$. Then for any $\delta > 0$ such that $n \ge \nu^{-2} \log(5/\delta)/2$, we have that with probability at least $1 - \delta$,*

$$R_{\pi_0}(\pi) = V(\overline{\pi}) - V(\pi_0) \le \overline{R}_{\pi_0}(\pi; \mathcal{U}) + 2\mathfrak{R}_n(\mathcal{F}) + \frac{C}{\nu} \sqrt{\frac{8 \log(5/\delta)}{n}} \qquad \forall \pi \in \mathcal{F} \tag{4}$$

In particular, if we let $\overline{\pi} = \overline{\pi}(\mathcal{F}, \mathcal{U}, \pi_0)$ be as in eq. (3) then eq. (4) holds for $\overline{\pi}$, which *minimizes* the right hand side. So, if the objective $\overline{R}_{\pi_0}(\pi; \mathcal{U})$ is negative, we are (almost) assured of getting some improvement on $\pi_0$. At the same time, so long as $\pi_0 \in \Pi$, the objective is necessarily non-positive, so we are also (almost) assured of doing no worse than $\pi_0$. Our guarantee of improvement holds, under well-specification, without requiring effect identification due to hidden confounding. Thus, Theorem 1 exactly captures the appeal of our approach.

### 3.3 Calibration of the uncertainty parameter $\Gamma$

In our framework, appropriate choice of $\Gamma$ is both important for ensuring that we avoid harm and will be context-dependent. The assumption that there exists a finite $\Gamma < \infty$ that satisfy eq. (2) is itself untestable, just like unconfoundedness (which corresponds to $\Gamma = 1$). Since we focus on enabling safe policy learning in domains where one errs toward safety in case of ignorance, if absolutely *nothing is known* then $\Gamma = \infty$ is the right choice and there is no hope for strictly safe improvement. However, practitioners generally have domain-level knowledge on the missing variables that may impact selection. This can guide the choice of $\Gamma < \infty$, which our method leverages to offer some improvement while ensuring safety. In particular, one way that the value of $\Gamma$ can be calibrated is by judging its value against the discrepancies in estimated propensities that are induced by omitting *observed* variables [10]. Then, determining a reasonable upper bound for $\Gamma$ can be phrased in terms of whether one thinks one has omitted a variable that could have increased or decreased the probability of treatment by as much as a particular observed variable. For example, a bound for $\Gamma$ can be implied by claiming one has not omitted a variable with as much impact on treatment as does, say, age, if age were observed. Additionally, when alternative outcome data is available, other approaches such as negative controls can be used to provide a lower bound for $\Gamma$ [16]. If one knows that the treatment does not have an effect on a particular outcome but one is observed in the data, then $\Gamma$ must be sufficiently large to invalidate that observed effect. These tools can be combined to derive a reasonable range for $\Gamma$ in practice. Since our focus is on safety, we suggest to err toward larger $\Gamma$.

## 4 Optimizing Robust Policies

We next discuss how to optimize the policy optimization problem in eq. (3). We focus on differentiable parametric policies, $\mathcal{F} = \{\pi(\,\cdot\,;\theta) : \theta \in \Theta\}$, such as logistic policies. We first discuss how to solve the worst-case regret subproblem for a fixed policy, which we will then use to develop our algorithm.

### 4.1 Dual Formulation of Worst-Case Regret

The minimization in eq. (3) for $\mathcal{U} = \mathcal{U}_n^\Gamma$ involves an inner supremum, namely $\overline{R}_{\pi_0}(\pi;\mathcal{U}_n^\Gamma)$. Moreover, this supremum over weights $W$ does not on the face of it appear to be convex. We next proceed to characterize this supremum, formulate it as a linear program, and, by dualizing it, provide an efficient procedure for finding the pessimal weights.

For compactness and generality, we address the optimization problem $\overline{Q}(r;\mathcal{U}_n^\Gamma)$ parameterized by an arbitrary reward vector $r \in \mathbb{R}^n$, where

$$\overline{Q}(r;\mathcal{U}) = \max_{W \in \mathcal{U}} \sum_{i=1}^n r_i W_i \big/ \sum_{i=1}^n W_i. \tag{5}$$

To recover $\overline{R}_{\pi_0}(\pi;\mathcal{U})$, we would simply set $r_i = 2(\pi(X_i) - \pi_0(X_i))T_i Y_i$. Since $\mathcal{U}_n^\Gamma$ involves only linear constraints on $W$, eq. (5) for $\mathcal{U} = \mathcal{U}_n^\Gamma$ is a *linear fractional program*. We can reformulate it as a linear program by applying the Charnes-Cooper transformation [5], requiring weights to sum to 1, and rescaling the pointwise bounds by a nonnegative scale factor $t$. We obtain the following equivalent linear program, where we let $w \in \mathbb{R}_+^n$ denote the normalized weights:

$$\overline{Q}(r;\mathcal{U}_n^\Gamma) = \max_{t,w \geq 0} \left\{ \sum_{i=1}^n r_i w_i : \ \sum_{i=1}^n w_i = 1; \ ta_i^\Gamma \leq w_i \leq tb_i^\Gamma, \forall\, i = 1,\ldots,n \right\} \tag{6}$$

The dual problem to eq. (6) has dual variables $\lambda \in \mathbb{R}$ for the weight normalization constraint and $u, v \in \mathbb{R}_+^n$ for the lower bound and upper bound constraints on weights, respectively, and is given by

$$\min_{u,v \geq 0, \lambda \in \mathbb{R}} \left\{ \lambda : \ b^\intercal v + a^\intercal u \geq 0, \ v_i - u_i + \lambda \geq r_i \ \forall\, i = 1...n \right\} \tag{7}$$

We use this to show that solving the adversarial subproblem requires only sorting the data and ternary search to optimize a unimodal function, generalizing the result of Aronow and Lee [1] for arbitrary pointwise bounds on the weights. Crucially, the algorithmically efficient solution will allow for faster subproblem solutions when optimizing our regret bound over policies in a given policy classes.

**Theorem 2** (Normalized optimization solution). *Let $(i)$ denote the ordering such that $r_{(1)} \leq r_{(2)} \leq \cdots \leq r_{(n)}$. Then, $\overline{Q}(r;\mathcal{U}_n^\Gamma) = \lambda(k^*)$, where $k^* = \inf\{k = 1,\ldots,n+1 : \lambda(k) < \lambda(k-1)\}$ and*

$$\lambda(k) = \frac{\sum_{i<k} a_{(i)}^\Gamma r_{(i)} + \sum_{i \geq k} b_{(i)}^\Gamma r_{(i)}}{\sum_{i<k} a_{(i)}^\Gamma + \sum_{i \geq k} b_{(i)}^\Gamma} \tag{8}$$

*Moreover, $\lambda(k)$ is a discrete concave unimodal function.*

Next we consider $\overline{Q}(r;\mathcal{U}_n^{\Gamma,\Lambda})$. Write an extended formulation for $\mathcal{U}_n^{\Gamma,\Lambda}$ using only linear constraints:

$$\mathcal{U}_n^{\Gamma,\Lambda} = \left\{ W \in \mathbb{R}_+^n : \exists d \text{ s.t. } \sum_{i=1}^n d_i \leq \Lambda, \ d_i \geq W_i - \hat{W}_i, \ d_i \geq \hat{W}_i - W_i, \ a_i^\Gamma \leq W_i \leq b_i^\Gamma \ \forall i \right\}$$

This immediately shows that $\overline{Q}(r;\mathcal{U}_n^{\Gamma,\Lambda})$ remains a fractional linear program. Indeed, letting $\overline{w}^0 = \sum_{i=1}^n \hat{W}_i$ a similar Charnes-Cooper transformation as used above with the additional normalization $d_i' = d_i t$ yields a non-fractional linear programming formulation:

$$\overline{Q}(r;\mathcal{U}_n^{\Gamma,\Lambda}) = \max_{d,w,t \geq 0} \left\{ \sum_{i=1}^n w_i r_i : \begin{array}{cc} \sum_i d_i' - \Lambda t \leq 0, & \sum_i w_i = 1, \ a_i t \leq w_i \leq b_i t \ \forall i \\ -d_i' \leq -w_i + \overline{w}_i^0 t, & -d_i' \leq w_i - \overline{w}_i^0 t, \ \forall i \end{array} \right\}$$

The corresponding dual problem is:

$$\min_{g,h,u,v,\nu \geq 0, \lambda \in \mathbb{R}} \left\{ \lambda : \begin{array}{c} v - u + g - h + \lambda \geq r, \quad v \geq g + h \\ -b^\intercal v + a^\intercal u - \Lambda \nu + g^\intercal \overline{w}^0 + h^\intercal \overline{w}^0 = 0 \end{array} \right\}$$

As $\overline{Q}(r;\mathcal{U}_n^{\Gamma,\Lambda})$ remains a linear program, we can easily solve it using off-the-shelf solvers.

## 4.2 Optimizing Parametric Policies

We next consider the case where $\mathcal{F} = \{\pi(\cdot;\theta) : \theta \in \Theta\}$, $\Theta$ is convex (usually $\Theta = \mathbb{R}^m$), and $\pi(x;\theta)$ is differentiable with respect to $\theta$. We suppose that $\nabla_\theta \pi(x;\theta)$ is given as an evaluation oracle. An example is logistic policies where $\pi(X;\alpha,\beta) = \sigma(\alpha + \beta^\intercal X)$ and $\Theta = \mathbb{R}^{d+1}$. Since $\sigma'(z) = \sigma(z)(1 - \sigma(z))$, evaluating derivatives is simple.

Our method follows a parametric optimization approach [26]. Note that $\overline{Q}(r;\mathcal{U})$ is convex in $r$ since it is a maximum over linear functions in $r$. Correspondingly, its subdifferential at $r$ is given by the argmax set:

$$\partial_r \overline{Q}(r;\mathcal{U}) = \left\{ \frac{W}{\sum_{i=1}^n W_i} : W \in \mathcal{U}, \ \frac{\sum_{i=1}^n W_i r_i}{\sum_{i=1}^n W_i} \geq \overline{Q}(r;\mathcal{U}). \right\}$$

If we set $r_i(\theta) = 2(\pi(X_i;\theta) - \pi_0(X_i))T_i Y_i$, so that $\overline{Q}(r;\mathcal{U}) = \overline{R}_{\pi_0}(\pi(\cdot;\theta);\mathcal{U})$, then $\frac{\partial r_i(\theta)}{\partial \theta_j} = 2T_i Y_i \frac{\partial \pi(X_i;\theta)}{\partial \theta_j}$. Although $F(\theta) := \overline{R}_{\pi_0}(\pi(\cdot;\theta);\mathcal{U})$ may not be convex in $\theta$, this suggests a subgradient descent approach. Let

$$g(\theta;W) = \nabla_\theta \frac{2\sum_{i=1}^n W_i(\pi(X_i;\theta) - \pi_0(X_i))T_i Y_i}{\sum_{i=1}^n W_i} = \frac{2\sum_{i=1}^n W_i T_i Y_i \nabla_\theta \pi(X_i;\theta)}{\sum_{i=1}^n W_i}.$$

Note that whenever $\partial_r \overline{Q}(r(\theta);\mathcal{U}) = \{W / \sum_{i=1}^n W_i\}$ is a singleton then $g(\theta;W)$ is in fact a gradient of $F(\theta)$.

At each step, our algorithm starts with a current value of $\theta$, then proceeds by finding the weights $W$ that realize $\overline{R}_{\pi_0}(\pi(\cdot;\theta))$ by using an efficient method as in the previous section, and then takes a step in the direction of $-g(\theta;W)$. Using this method, we can optimize policies over both the unbudgeted uncertainty set $\mathcal{U}_n^\Gamma$ and the budgeted uncertainty set $\mathcal{U}_n^{\Gamma,\Lambda}$. Because descent is not always guaranteed at each step, at the end, we return the value of $\theta$ that corresponds to the best objective value seen so far. Our method is summarized in Alg. 1.

## 5 Experiments

**Simulated data.** We first consider a simple linear model specification demonstrating the possible effects of significant confounding on inverse-propensity weighted estimators.

$$\xi \sim \text{Bern}(1/2), \quad X \sim N(\mu_x, I_5), \quad U = \mathbb{I}[Y_i(1) < Y_i(-1)]$$

$$Y(t) = \beta_0^\intercal x + 1/2(t+1)\beta_{treat}^\intercal x + \alpha 1/2(t+1) + \eta \xi t + \omega \xi + \epsilon$$

The constant treatment effect is 2.5 with the linear interaction $\beta_{treat} = [-1.5, 1, -1.5, 1, 0.5]$. The covariate mean is $\mu_x = [-1, .5, -1, 0, -1]$. The noise term $\xi$ affects outcomes with coefficients $\eta = -2, \omega = 1$, in addition to a uniform noise term $\epsilon \sim N(0,1)$. We let the nominal propensities be logistic in $X$, $\hat{e}(X_i) = \sigma(\beta^\intercal x)$ with $\beta = [0, .75, -.5, 0, -1, 0]$, and we generate $T_i$ according to the true propensities, which we set to $e(X_i, U_i) = \frac{4 + 5U + \hat{e}(X_i)(2 - 5U)}{6\hat{e}(X_i)}$.

We compare the policies learned by a variety of methods. We consider two commonplace standard methods that assume unconfoundedness: the logistic policy minimizing the IPW estimate with

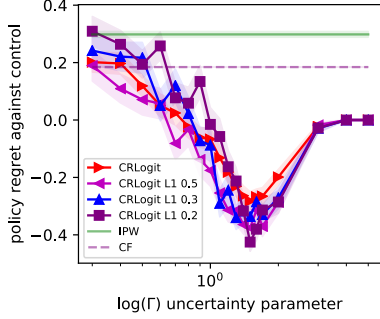

Figure 1: Out of sample policy performance on synthetic data, where the true generating $\log(\Gamma)^* = 1.5$.

Algorithm 1: Parametric Subgradient Method

1: Input: step size $\eta_0$, step-schedule exponent $\kappa \in (0, 1]$, initial iterate $\theta_0$, number of iterations $N$
2: **for** $t = 0, \ldots, N - 1$ **do**:
3:     $\eta_t \leftarrow \eta_0 t^{-\kappa}$            $\triangleright$ Update step size
4:     $\ell_t \leftarrow \max\limits_{W \in \mathcal{U}} \frac{2\sum_{i=1}^{n} W_i(\pi(X_i;\theta_t) - \pi_0(X_i))T_i Y_i}{\sum_{i=1}^{n} W_i}$
5:     $W \leftarrow \arg\max\limits_{W \in \mathcal{U}} \frac{2\sum_{i=1}^{n} W_i(\pi(X_i;\theta_t) - \pi_0(X_i))T_i Y_i}{\sum_{i=1}^{n} W_i}$
6:     $\theta_{t+1} \leftarrow \text{Projection}_{\Theta}(\theta_t - \eta_t g(\theta_t; W))$
    **return** $\theta_{\arg\min_t \ell_t}$

nominal propensities[1] and the direct comparison policy gotten by estimating CATE using causal forests and comparing it to zero [CF; 31]. We compare these to our methods with a never-treat baseline policy $\pi_0(x) = 0$: our robust logistic policy using the unbudgeted uncertainty set, our robust logistic policy using the budgeted uncertainty set and multipliers $\rho = 0.5, 0.3, 0.2$. For each of these we vary the parameter $\Gamma$ in $\{0.3, 0.4, \ldots, 1.6, 1.7, 2, 3, 4, 5\}$. The causal forest policy achieves slightly better regret than the IPW policy, but remains confounded. By construction, for $\log(\Gamma)$ very small (left end of plot), the confounding-robust approach tracks IPW with the nominal propensities and incurs some regret relative to control. When we add robustness, our policies achieve substantial improvements. As $\log(\Gamma)$ increases, the learned robust logistic policies are able to achieve negative regret, meaning we improve upon $\pi_0$. As $\log(\Gamma)$ grows very large (right end of plot), we are very robust to any size of confounding and almost always default to $\pi_0$ as a policy that ensures never doing worse and our true regret converges to 0. Even in this extreme example of confounding where the true propensities achieve the odds-ratio bounds, the budgeted version is able to attain similar improvements to the unbudgeted version for $\rho = 0.3, 0.2$, and uniformly better improvements for $\rho = 0.5$. These improvements are relatively insensitive to the exact value of $\rho$ and the budgeted version is able to achieve improvement even when the budgeted uncertainty set is misspecified. The best improvements for the parametric policies are achieved at $\log(\Gamma) = 1.5$, consistent with the model specification.

**Assessment with Clinical Data: International Stroke Trial.**

We build an evaluation framework for our methods from real-world data, where the counterfactuals are not known, by simulating confounded selection into a training dataset, and estimating out-of-sample policy regret on a held-out "test set" from the completely randomized controlled trial. We study the International Stroke Trial (IST), restricting attention to two treatment arms from the original factorial design: the treatment arm of both aspirin and heparin (high dose) ($T = 1$) vs. only aspirin ($T = -1$) treatment arms, numbering 7233 cases with $\Pr[T = 1] = \frac{1}{3}$ [8]. We defer some details about the dataset to Appendix C. Findings from the study suggest clear reduction in adverse events (recurrent stroke or death) from aspirin, whereas heparin efficacy is inconclusive since small (non-significant) benefit on rates of death at 6 months was offset by greater incidence of other adverse events such as hemorrhage or cranial bleeding. We construct an evaluation framework from the dataset by first sampling a split into a training set $S_{\text{train}}$ and a held-out test set $S_{\text{test}}$, and subsampling a final set of initial patients, whose data is then used to train treatment assignment policies. We generate nominal *selection* probabilities into the final training set, letting $Z = 1$ denote inclusion, as $\Pr[Z = 1 \mid X_{\text{age}}] = 0.6 + 0.2X_{\text{age}}$, where $X_{\text{age}} \in [0, 1]$ is rescaled. Then the nominal propensities of treatment assignment in the final training set are $\Pr[Z = 1, T = 1 \mid X] = 0.2 + 0.1X_{\text{age}}$. We introduce confounding by censoring the treated patients with the worst 10% of outcomes, and the 10% best patients in the control group.

The original trial measured a set of clinical outcomes including death, stroke recurrence, adverse side effect, and full recovery at six months: we scalarize these outcomes as a composite loss function. A

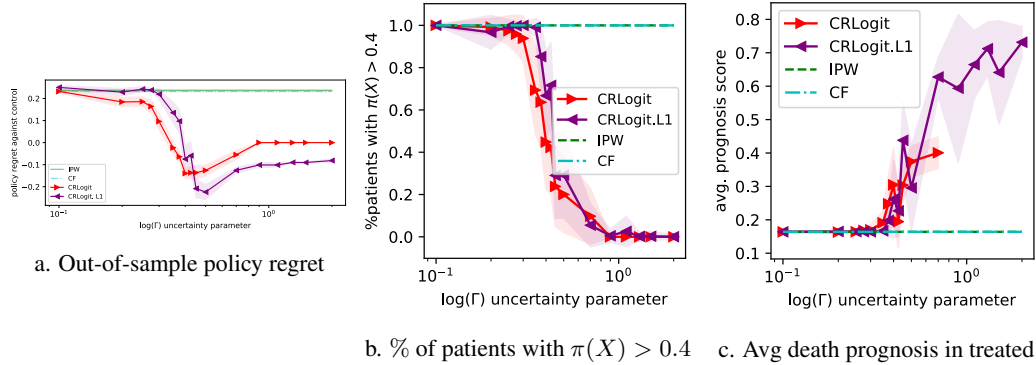

a. Out-of-sample policy regret

b. % of patients with $\pi(X) > 0.4$    c. Avg death prognosis in treated

Figure 2: Comparison of policy performance on clinical trial (IST) data as $\Gamma$ increases

difference-in-means estimate of the ATE for the composite score in full data is significant at $0.13$, suggesting that heparin is overall harmful. Without access to the true counterfactual outcomes for patients, our oracle estimates are IPW-based estimates from the held-out RCT data with probabilities of treatment assignment as $p_{-1} = \frac{2}{3}$ and $p_1 = \frac{1}{3}$. We use an out-of-sample Horvitz-Thompson estimate of policy regret relative to $\pi_0(x) = 0$ based on the held-out dataset $S_{\text{test}}$, $R_{\pi_0}^{\text{test}}(\pi) = \frac{1}{|S_{\text{test}}|}\sum_{i \in S_{\text{test}}} Y_i T_i \pi(X_i) \frac{1}{p_{T_i}}$. In Fig. 2a, we evaluate on 10 draws from the dataset, comparing our policies against the vanilla IPW estimator $\sum_i \frac{Y_i \Pr[\pi_i = T_i]}{\Pr[T = T_i]}$ with a probabilistic policy, and assigning based on the sign of the CATE prediction from causal forests [31]. The selected datasets average a size of $n_{train} = 2430$. We evaluate logistic parametric policies (CRLogit) and budgeted (CRLogit.L1) with $\rho = 0.5$. For the parametric policies, we optimize with the same parameters as earlier. We evaluate $\log(\Gamma) = 0.1, 0.2$, every $0.025$ between $0.25$ and $0.45$, every $0.2$ between $\log(\Gamma) = 0.5, 1.5$ and $\Gamma = 2$. For small values of $\log(\Gamma)$, our methods perform similarly as IPW. As $\log(\Gamma)$ increases, our methods achieve policy improvement, though the L1-budgeted method (CRLogit.L1) achieves worse performance. For $\log(\Gamma) > 0.9$, the robust policy essentially learns the all-control policy; our finite-sample regret estimator simply indicates good regret for a neglible number of patients (5-6).

In Figs. 2b-2c, we study the behavior of the robust policies. The IST trial recorded a prognosis score of probability of death at 6 months for patients, using an externally validated model, which we do not include in the training data, but use to assess the validity of our robust policy. In Fig. 2c, we consider the average prognosis score of death for among patients treated with $\pi(X) > 0.4$. In Fig. 2b, for $\log(\Gamma) \in [0.3, 0.5]$, the policy considers treating $1 - 20\%$ of patients and the subsequent average prognosis score of the population under consideration increases, indicating that the policy is learning and treating on appropriate indicators of severity from the available covariates. For $\log(\Gamma) > 0.9$, the noise in the prognosis score is due to the small treated subgroups (while the unbudgeted policy does not learn a policy that improves upon control, so we default to control and truncate the plot).

Our learned policies suggest that improvements from heparin may be seen in the highest-risk patients, consistent with the findings of [2], a systematic review comparing anticoagulants such as heparin against aspirin. They conclude from a study of a number of trials, including IST, that heparin provides little therapeutic benefit, with the caveat that the trial evidence base is lacking for the highest-risk patients where heparin may be of benefit. Thus, our robust method appropriately treats those, and only those, who stand to benefit from the more aggressive treatment regime.

# 6 Conclusion

We developed a framework for estimating and optimizing for robust policy improvement, which optimizes the minimax regret of a candidate personalized decision policy against a baseline policy. We optimize over uncertainty sets centered at the nominal propensities, and leverage the optimization structure of normalized estimators to perform policy optimization efficiently by subgradient descent on the robust risk. Assessments on synthetic and clinical data demonstrate the benefits of robust policy improvement.

**Acknowledgments**

This material is based upon work supported by the National Science Foundation under Grant No. 1656996. Angela Zhou is supported through the National Defense Science & Engineering Graduate Fellowship Program.

## Footnotes

[1]We also tried the self-normalized variant of Swaminathan and Joachims [27] and report the results in Sec. B in the appendix.

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
