[Supplementary Material]

# A  Proofs

*Proof of Theorem 1.* Let $Z = \frac{1}{n}\sum_{i=1}^n W_i^*$. Then

$$\mathcal{D} = \sup_{\pi \in \mathcal{F}} \left| \hat{V}^*(\pi) - V(\pi) \right| \leq \frac{2}{Z} \sup_{\pi \in \mathcal{F}} \left| \frac{1}{n}\sum_{i=1}^n \frac{1}{2} W_i^*(1 + T_i(2\pi(X_i) - 1))Y_i - V(\pi) \right| + \frac{C\left|2 - Z\right|}{Z}$$

Since $W_i^* \in [0, \nu^{-1}]$ and $\mathbb{E}W_i^* = 2$, Hoeffding's inequality gives

$$\mathbb{P}\left( |Z - 2| \geq \epsilon \right) \leq 2\exp(-2\epsilon^2\nu^2 n).$$

Let

$$\mathcal{D}' = \sup_{\pi \in \mathcal{F}} \left| \frac{1}{n}\sum_{i=1}^n \frac{1}{2} W_i^*(1 + T_i(2\pi(X_i) - 1))Y_i - V(\pi) \right|$$

and note that since $|1 + T_i(2\pi(X_i) - 1)| \leq 1$ we have that $\mathcal{D}'$ satisfies bounded differences with constant $2C/(n\nu)$. Hence, McDiarmid's inequality gives

$$\mathbb{P}\left( \mathcal{D}' - \mathbb{E}\mathcal{D}' \geq \epsilon \right) \leq \exp(-\epsilon^2\nu^2 C^{-2}n/2).$$

Next, because $\mathbb{E}\frac{1}{2}W_i^*(1 + T_i(2\pi(X_i) - 1))Y_i = V(\pi)$, a standard symmetrization argument gives that

$$\mathbb{E}\mathcal{D}' \leq \mathbb{E}\frac{1}{2^n}\sum_{\epsilon \in \{-1,+1\}^n} \sup_{\pi \in \mathcal{F}} \left| \frac{1}{n}\sum_{i=1}^n \epsilon_i W_i^*(1 + T_i(2\pi(X_i) - 1))Y_i \right|$$

Further, by the Rademacher comparison lemma [14, Thm. 4.12], we get that

$$\mathbb{E}\mathcal{D}' \leq 2C\nu^{-1}\mathbb{E}\mathfrak{R}_n(\mathcal{F}).$$

Next, $\mathfrak{R}_n(\mathcal{F})$ satisfies bounded differences with constant $2/n$ so McDiarmid's inequality gives

$$\mathbb{P}\left( \mathbb{E}\mathfrak{R}_n(\mathcal{F}) - \mathfrak{R}_n(\mathcal{F}) \geq \epsilon \right) \leq \exp(-\epsilon^2 n/2).$$

Finally, Hoeffding's inequality gives that

$$\mathbb{P}\left( \hat{V}^*(\pi_0) - V(\pi_0) \geq \epsilon \right) \leq \exp(-\epsilon^2\nu^2 C^{-2}n/2).$$

Combining, we get that with probability at least $1 - p_1 - p_2 - p_3 - p_4$, assuming that $\frac{1}{\nu}\sqrt{\frac{\log(2/p_1)}{2n}} \leq 1$, we have that $\sup_{\pi \in \mathcal{F}}(R_{\pi_0}(\pi) - \hat{R}_{\pi_0}^*(\pi))$ is bounded above by

$$2\frac{C}{\nu}\sqrt{\frac{2\log(1/p_2)}{n}} + 2\mathfrak{R}_n(\mathcal{F}) + 2\sqrt{\frac{2\log(1/p_3)}{n}} + \frac{C}{\nu}\sqrt{\frac{\log(2/p_1)}{2n}} + \frac{C}{\nu}\sqrt{\frac{2\log(1/p_4)}{n}}.$$

Letting $p_2 = p_3 = p_4 = \delta/5$ and $p_1 = 2\delta/5$, the above is bounded by $2\mathfrak{R}_n(\mathcal{F}) + \frac{2C}{\nu}\sqrt{\frac{2\log(5/\delta)}{n}}$ so long as $n \geq \nu^{-2}\log(5/\delta)/2$. The proof is completed by noting that by assumption of true weights being inside $\mathcal{U}$ we get that $\hat{R}_{\pi_0}^*(\pi) \leq \overline{R}_{\pi_0}(\pi; \mathcal{U})$. $\qquad\square$

*Proof.* Proof of the equivalence of programs (5) and (6). We can easily verify that a feasible solution for one problem is feasible for the other: for a feasible solution $W$ to (FP), we can generate a feasible solution to (LP) as $w_i = \frac{W_i}{\sum_i W_i}, t = \frac{1}{\sum_i W_i}$ with the same objective value. In the other direction, we can generate a feasible solution to (6) from a feasible fractional program (5) solution $W, t$ if we take $w_i = w_i t$. This solution has the same objective value since $\sum_i w_i = 1$. $\qquad\square$

*Proof.* Proof of Thm. 2. We analyze the program using complementary slackness, which will yield an algorithm for finding a solution that generalizes that of Aronow and Lee [1].

At optimality only one of the primal weight bound constraints, (for nontrivial bounds $a^\Gamma < b^\Gamma$), $w_i \leq tb_i^\Gamma$ or $ta_i^\Gamma \leq w_i$ will be tight. For the nonbinding primal constraints, at the optimal solution, by complementary slackness at most one of $u_i$ or $v_i$ is nonzero. Furthermore, $t \neq 0$ as $t = 0$ is infeasible. The constraint $\sum_i -b_i v_i + a_i u_i \geq 0$ is active at optimality, otherwise there exists smaller

yet feasible $\lambda$ that achieves a lower objective of the dual program. So the optimal solution to the dual will satisfy:

$$\min \lambda$$
$$\text{s.t. } \lambda \geq r_i + u_i - v_i, \ \forall i \in 1, \ldots, n$$
$$\sum_i -b_i^\Gamma v_i + a_i^\Gamma u_i = 0$$

By non-negativity of $u_i, v_i$, note that $u_i > 0$ if $r_i < \lambda$ and $v_i > 0$ if $r_i > \lambda$ such that $u_i = \max(0, \lambda - r_i)$ and $v_i = \max(0, r_i - \lambda)$. Additionally, feasible objective values satisfy $\lambda \leq \max_i Y_i$ and $\lambda \geq \min_i Y_i$. Let $(k)$ denote the $k$th index of the increasing order statistics, an ordering where $r_{(1)} \leq r_{(2)} \leq \cdots \leq r_{(n)}$. Then at optimality, there exists some index $(k)$ where $Y_{(k)} < \lambda \leq Y_{(k+1)}$. We can subsitute in the solution from the binding constraints $\lambda = r_i + u_i - v_i$ and obtain the following equality which holds at optimality:

$$\sum_{i:(i)<(k)} a_{(i)}^\Gamma(\lambda - r_{(i)}) - \sum_{i:(i)\geq(k)} b_{(i)}^\Gamma(r_{(i)} - \lambda) = 0$$

$$\sum_{i:(i)<(k)} a_{(i)}^\Gamma \lambda + \lambda \sum_{(i)>(k)} b_{(i)}^\Gamma = \sum_{i:(i)\geq(k)} b_{(i)}^\Gamma r_{(i)} + \sum_{i:(i)<(k)} a_{(i)}^\Gamma r_{(i)}$$

$$\lambda_{(k)} = \frac{\sum_{i:(i)<(k)} a_{(i)}^\Gamma r_{(i)} + \sum_{i:(i)\geq(k)} b_{(i)}^\Gamma r_{(i)}}{\sum_{i:(i)<(k)} a_{(i)}^\Gamma + \sum_{i:(i)\geq(k)} b_{(i)}}$$

Therefore, we only need to check the possible objective values $\lambda_{(k)}$ for $k = 1, \ldots, n$. The primal solution is easily recovered from the dual solution: for $r_{(i)}$, take $w_{(i)} = \frac{a_{(i)}^\Gamma \mathbb{I}\{(i)\leq k\} + b_{(i)}^\Gamma \mathbb{I}\{(i)>k\}}{\sum_{i:(i)<(k)} a_{(i)}^\Gamma + \sum_{i:(i)\geq(k)} b_{(i)}^\Gamma}$ and $t = \sum_{i:(i)<(k)} a_{(i)}^\Gamma + \sum_{i:(i)\geq(k)} b_{(i)}^\Gamma$. Consider the parametric restriction of the primal program, where it is parametrized by the sum of weights $t$: the value function is concave in $t$ and concave in the discrete restriction of $t$ to the values it takes at the solutions of $\lambda_{(k)}$, $t_{(k)}$, and $t_{(k)}$ is increasing in $k$. So the optimal such $\lambda$ occurs with the order statistic threshold at $(k)$ for $k^* = \inf\{k = 1, \ldots, n+1 : \lambda(k+1) < \lambda(k)\}$. $\qquad\square$

## B   Additional Experimental Details

**Comparison to SNPOEM:**   In the simulated example in Sec. 5, we additionally assess the performance of a self-normalized counterfactual policy maximizer, SNPOEM [27]. This approach adds both a variance regularization and a self-normalization, both of which strongly bias the learned policy toward the logging policy. While this has merit, it results in spuriously inconsistent results across the problems we consider, where in one it has reasonable results and in another much worse results than any other standard method. Specifically, SNPOEM achieves a mean policy value of 0.82 (SD 0.04) in the simulated example and 0.04 (SD 0.029) in the IST example. Because this additional layer of complexity only confuses the comparisons and the main focus on the contrast between ignoring and accounting for unconfoundedness, we omit these results from the main text.

## C   IST data details

The International Stroke Trial assessed the clinical effectiveness of aspirin, subcutaneous heparin, both, or neither among 19435 patients with acute ischaemic stroke in a factorial design.

Our scalarized composite score is:

$$Y = 2\mathbb{I}[\text{death}] + \mathbb{I}[\text{recurrent stroke}] + 0.5\mathbb{I}[\text{pulmonary embolism or intracranial bleeding}]$$
$$+ 0.5\mathbb{I}[\text{other side effects}] - 2\mathbb{I}[\text{full recovery at 6 months}] - \mathbb{I}[\text{discharge within 14 days}]$$

Covariates available at the time of randomization used by us to train data include age, indicators of conscious state, sex, blood pressure, and factors describing clinical assessments such as visible

infarct on CT, face decifit, arm/face deficit, leg/foot deficit, dysphasia, hemianopia, stroke subtype, visuospatial disorder, brainstem/cerebellar signs, and other deficit. We construct one-hot encodings of the categorical variables and train policies on 28 covariates after the dummy encoding.