[Reviews · NeurIPS 2018]

Reviewer 1



This paper studies the conditions under which IPW-based policy may fail due to the existence of latent confounders, and then provides methods for obtaining a robust policy through optimization of the worst-case regret (which is motivated by the sensitivity analysis literature). Given a user-set \Gamma, the bound of the odds ratio (as in the sensitivity-analysis), and the assumption of e(x,u) (propensity) as a logistic model, the authors derive bounds over the true inverse propensity. This leads to the bound on an empirical regret.  Overall, I enjoyed the paper; the writing is clear and well organized. It successfully conveys the idea of the authors. However, as the paper currently stands, there are still important issues that are somewhat unclear and/or not present, which I list below.  Q1. Setting parameters  Given erlyspecif bounds — \Gamma and \Lambda, one can see improvements on the policy. Then, it is natural to ask how the user should think about and decide the values of \Gamma and-or \Lambda. In sensitivity analysis, researchers can try different values of \Gamma to ‘investigate’ how the results of a study would vary. However, the purpose of this paper is to ‘replace’ the given policy w/ the resulting robust-policy w.r.t. \Gamma, so it's not entirely clear to me how to perform such a thought experiment. Please, elaborate on this issue.  Q2. Data-driven Confounding Robust Policy Improvement The sensitivity parameter is set by the user, leading to a bound on the odds ratio, which eventually will constrain the policy value, or the causal effect of the treatment. There is, however, an entirely data-driven approach for constraining the experimental distribution, which is the final target of the analysis. As I understand the paper, it tries to bound the PS since the same is not identifiable when unobserved confounding is present. Still, the real goal of learning the PS, in the first place, is to estimate the Y_T distribution.  If one can bound Y_T directly from P(X, T, Y), it seems that asking for the user input may be unnecessary (since one could immediately get a "data-driven" gamma). The idea of bounding E[Y_T] directly from data (under confounding) seems to be the route taken by (ZB, 2017) (see also Ch. 8, Causality, 2000). Given that both approaches appear concerned with optimizing online learning, how are they related? We further note that it's possible that the user parameters may lead to bounds that are less informative (larger) than the ones coming directly from the observational (logged) data. Please, can you elaborate on this point?  Q3. Empirical Evaluation  a. If the data is obtained without confounding, what would be the performance of the robust policy returned by the algorithm? Even though the paper demonstrated how robust policy could be worsened with unnecessarily large \Gamma, still there is no natural way, as far as I can see, to tell what \Gamma should be.  b. How about the robustness against the violation of the logistic model assumption of e(x,u)? In such a case, is there any natural interpretation of \Gamma?  c. Why is the IPW worse than RIC family when \Gamma=1? (Figure 2 — referenced as Fig 1a) Is it because the used IPW is a vanilla implementation?  It would be crucial to see results of weight-clipped IPW or weight-normalized IPW.  d. If the data is generated from a different distribution, is it possible for IPW to perform better than RIC* w/ \Gamma=1? e. Similarly, the performance gain due to the use of \Gamma should be compared to \Gamma=1, not to a vanilla IPW. Please, justify and clarify the meaning of RIC performance at \Gamma=1. Isn’t it assuming that there is no confounder? The above questions are similarly applied to the experiments with IST data. RIC*s appear to be improved as \Gamma increases (Fig 2a, 2b). However, considering \Gamma=1, we only see a degradation of the performance with smaller \Gammas. It is somewhat hard to assess where the performance gain is coming from. Summary:  - Clarity: The paper is overall well-written with a right balance between text and math. - Originality: It is original in that it provides a new efficient method to yield a robust policy by optimizing worst-case regret, under the assumptions that the user can provide bounds. This judgment is contingent on how this approach relates (generalizes? subsumes?) the data-driven approach discussed above. - Significance: Without knowing how to set \Gamma, the applicability of the method may be limited, and the improvement guarantee may be somewhat illusive. The baseline should be replaced to accurately assess how the proposed methods work w/ different \Gammas.  - Quality: Overall okay, with good mathematical details but without strong empirical evidence and not very accurate literature review and comparison. Minor comments: for for (Line 173) an uncertainty (Line 9) There are two figure 2s without figure 1. Figure 2 (a), (b), (c) not marked. Reference:  Zhang, J. and Bareinboim, E. Transfer Learning in Multi-Armed Bandits: A Causal Approach. In Proceedings of the 26th International Joint Conference on Artificial Intelligence (IJCAI), 1340-1346, 2017.   Pearl, J. Causality: Models, Reasoning and Inference. Cambridge University Press, 2000. --- Post-rebuttal  We believe the authors addressed most of the issues and updated my grades accordingly. We would like to highlight that we expect the authors doing a better job in the final version of the paper regarding the two central points:  1. Gamma is the critical parameter that the whole analysis hinges on, so it should have more emphasis an explanation on how, in any cognitively meaningful way, users can conceive setting these parameters to some value.  2. I don’t think the answer regarding Q2 is accurate. The paper should acknowledge that there’s a tradeoff between knowledge an inferential power, and they are not proposing anything magical here. If the user wants to be entirely data-driven, they should go with the work of ZB'17 (or some obvious/natural generalization), while if they do have the substantive knowledge needed to set \Gamma (whatever this could be), they can consider the approach proposed in the paper. These points should be front and center in the paper since it currently conveys the image that nothing was done before and that \Gamma is a somewhat "magical" parameter. Further, it seems the results are obtained without any effort and there isn't a more data-driven alternatively to their method. Furthermore, the work on Manski and other folks are great for bounding in the offline setting, but the desirable property about the setting discussed is coupling the knowledge from one policy on another and proving that it can indeed help in practice.  Still, I think the work proposed here is nice and should, therefore, be accepted.

Reviewer 2



Most methods used for estimating causal quantities from observational data relf on unconfoundedness i.e. there exist no unobserved confounders between treatment and outcome. In reality however, this assumption is rarely satisfied. Assuming that unconfoundedness holds (when it doesn’t) can create major complications in sensitive applications such as medicine. This paper develops a framework for policy improvement that works well in the presence of confounding.

Reviewer 3



This paper conducts a sensitivity analysis for learning optimal treatment regimes. Unlike traditional analysis in optimal treatment regimes, it allows for unmeasured confounding to a certain degree characterized by a sensitivity parameter \Gamma, in the same way as defined by Rosenbaum 2002. To learn the optimal robust policy, they turn the optimization problem into a linear program, which can be solved efficiently. Overall I feel this is an interesting direction. I have the following comments. 1. Why do you choose to optimize the regret function, instead of the loss function \hat{V}^*(\pi) directly? The latter seems to be more relevant from a patient’s perspective. 2. To follow up on that, although this is counterintuitive, mathematically I think there is no guarantee that the two solutions mentioned in my first point coincide with each other. I think this is a key point of this paper, so it may be worth having a toy example illustrating the difference, and explain why the proposed estimand is better. 3. Suppose that the standard care is all control. Consider the following approach: for each parameter value X_i, given \Gamma, compute the bound for ITE(X_i). If the lower bound is greater than 0, set \pi(X_i) = 1; if the upper bound is smaller than 0, set \pi(X_i) = 0; otherwise let \pi(X_i) = ½. Does this approach always give you the optimal one over an unrestricted policy class? 4. I liked the idea of budgeted uncertainty sets, and it looks like it is more general than the unbudgeted one (i.e. assume constant \Gamma for all). However, it looks like that it is not doing well in the experiments. Can you provide some intuition into why this is the case? 5. Is it always desirable to include the control/baseline policy in the policy class? If so, I would expect the regret function to be always non-positive, which is contrary to what I see in the experiments. 6. In the clinical data application, you have a composite outcome consisting of death and other secondary outcomes, such as stroke. However, it is commonly believed that stroke is undefined for dead people, and comparison for stroke among survived people is subject to selection bias. This problem is sometimes known as truncation-by-death in the literature. I am wondering how did you get around this problem in the application. 7. Do you have any comments on the uncertainty region around the optimal policy? 8. Does the approach work if U is multi-dimensional? Minor Comments: 1. Figure 1 is numbered as Figure 2 2. Figure 2: Can you make the scale and range of x-axis consistent across three plots? 3. In the application, it was said that findings from the study suggest clear reduction in adverse events from aspirin. I am wondering how this is possible, given that both treatment arms are given aspirin. 4. Page 7 line 291: what is \delta_age 1/3 ?